Microbiology
Spectrum

# Distinctive structure of endophytic microbial communities in two species of wild and cultivated rice

Lingyun Lei,[1,2] Xingyi Li,[1] Zixuan Xiong,[1] Jinlu Li,[1] Li Liu,[1] Ling Chen,[1] Qiaofang Zhong,[1] Hui Jiang,[1] Zaiquan Cheng,[1] Suqin Xiao[1]

**ABSTRACT** Endophytic microbial communities play an important role in plant development, nutrient acquisition, and oxidative stress tolerance. *Oryza officinalis* and *Oryza meyeriana* are unique wild rice varieties in China with many high-quality resistance genes, rich endophyte diversity, and potential resources for sustainable agriculture. In the present study, the endophytic microbial community structures of *O. officinalis*, *O. meyeriana*, and cultivated rice were compared using metagenomic sequencing. *Dechloromonas*, *Salmonella*, *Klebsiella*, and *Listeria* were the core microbial groups in wild and cultivated rice. The relative abundances of *Ligilactobacillus*, *Escherichia*, and *Bradyrhizobium* in *O. meyeriana* were higher than those in cultivated rice. The relative abundances of *Listeria*, *Acinetobacter*, *Escherichia*, and *Dechloromonas* in *O. officinalis* were also higher. Compared to that of cultivated rice, the microbiota of wild rice had a more complex and stable community network. At the pathway level 2 based on the Kyoto Encyclopedia of Genes and Genomes classification system, the relative abundance of metabolic categories was dominant. Most pathways showed that the *O. officinalis* relative abundance was higher than those of the other two species. Our study revealed differences in the leaf endophyte community structure and function between wild and cultivated rice in the same habitat, demonstrating the potential of wild rice in recruiting specific microorganisms to improve crop performance and promote safe and sustainable food production.

**IMPORTANCE** Under the current climate change environment, plant-beneficial endophytes are of great significance for promoting food production. Modern cultivars may have lost many beneficial endophytes compared to their ancestors. However, relatively few studies have been conducted on the community structure and function of modern ancestral crop endophytes. In this study, the composition and function of the microbial communities of two wild rice species were analyzed; the differences between them and cultivated rice were determined; and the patterns of microbial interactions and their core microbiomes were determined. Our findings can aid in the exploration of the beneficial endophytes in wild rice and use them to improve crop stress resistance and sustainability. These results provide relevant insights into the role of endophytes in the mechanism of high-stress resistance in wild rice.

**KEYWORDS** *Oryza officinalis*, *Oryza meyeriana*, endophytes, community structure and function, difference analysis, co-occurrence network

Plant endophytes, a diverse group comprising fungi, bacteria, and actinomycetes, colonize plant tissues either intercellularly or intracellularly. Although several endophytic pathogens are found (1), the majority establish neutral or beneficial associations without causing host damage or symptoms (2–4). Confirmed endophytic functions include biological nitrogen fixation (5), phytohormone production (6), siderophore biosynthesis (7), antibacterial metabolite secretion (8), and induction of

**Peer Reviewers** Malick Bill, USDA-ARS Northern Crop Science Laboratory, Fargo, North Dakota, USA; Jacob A. Heil, Boise State University, Boise, Idaho, USA

Address correspondence to Zaiquan Cheng, czquan-99@163.com, or Suqin Xiao, xiaosuqin227@126.com.

The authors declare no conflict of interest.

See the funding table on p. 13.

systemic plant defense (9). In addition, some endophytic bacteria facilitate metal detoxification in host plants, contributing to phytoremediation of contaminated soils (10). The spatial distribution of endophytes exhibits tissue-specific patterns: endophytic fungi are more abundant in the leaves, sheaths, and seeds than in the roots (11). This colonization is modulated by multiple factors, including the plant genotype, environmental variables, growth stage, and interactions with other organisms. The functional versatility of endophytes stems from their taxonomic diversity, with microorganisms universally affecting plant development under natural conditions. As highlighted by Berg et al. in 2016, endophytes play critical roles in modulating host physiology and stress responses through complex microbiome interactions (12).

Rice (*Oryza sativa* L.) is the staple food and main nutrition source for 2.5–3.5 billion people globally, and its demand is growing in some low-income countries (13, 14). The stable rice yield is limited by various biotic and abiotic stresses. With the increase in world population, the current rice production growth rate will not meet global food supply needs (15, 16). Researchers are focusing on the community structure of rice-associated microbiomes to develop a new, stable, and sustainable rice production. Wild rice is the ancestor of rice and possesses excellent traits, such as disease and insect resistance and high yield under long-term natural field growth conditions. In addition, it is an important germplasm resource for improving cultivated rice (17). In China, three *Oryza* species are used: *Oryza officinalis*, *Oryza meyeriana*, and *Oryza rufipogon*. *O. officinalis* has a diploid CC genome and several ideal agronomic characteristics similar to those of C4 plants, such as a strong growth potential, high photosynthetic efficiency, large biomass accumulation, and strong resistance to biotic and abiotic stresses (18, 19). *O. meyeriana*, with its GG genome, is the largest known *Oryza* species characterized by high resistance to bacterial blight, insect and drought resistance, and shade tolerance (20). Wild rice and host endophytic bacteria are believed to have co-evolved to benefit host health through growth promotion, nutrient absorption, and stress resistance (21, 22). Compared with cultivated rice, wild rice endophytes are more valuable and diverse microbial biological resources (23). In other words, during domestication, crops may lose genetic diversity as well as some host-specific endophytes, as it has been confirmed in wheat (24). Therefore, the analysis of the community structure and function of wild rice endophytes aids in elucidating the interaction mechanism between hosts and endophytes and provides a theoretical basis for using beneficial endophytes to improve rice production in the future.

Compared with cultivated varieties, wild varieties have higher beneficial endophytic content and lower pathogen content (25). In an early study, the evolutionary relationship in wheat-microorganism interactions was first revealed (26), and ancient wheat cultivars are colonized by phylogenetically diverse rhizobacterial isolates, whereas the rhizosphere of modern cultivars is dominated by fast-growing *Proteobacteria*. The *O. rufipogon* rhizosphere and phyllosphere microbial communities have been preliminarily characterized (23, 27). When African or Asian wild rice varieties were crossed with cultivated rice varieties, a greater number of root endophytic fungi was observed in the first generation compared to that of the cultivated parents used for the hybridization (28). However, to date, the microbial diversity, community structure, and microbial interactions of *O. officinalis* and *O. meyeriana* leaf endophytes remain unclear.

In this study, the leaves of two types of wild rice (*O. officinalis* and *O. meyeriana*) and Nipponbare used as a control were collected from Yunnan Province, China, and the endophytic bacterial communities in their leaves were characterized using metagenomic sequencing. The aims of this study were to (i) identify the leaf endophytic microbial composition of each type of rice and the core microflora of the three types of rice through species annotation, (ii) compare the differences in endophytic microbial communities between wild and cultivated rice, (iii) analyze the functional composition of each type of rice and compare the differences through functional annotation, and (iv) construct a molecular network interaction model of different types of rice and compare the similarities and differences between them. Our results systematically compared

the endophyte communities of two types of wild rice and cultivated rice, providing a theoretical basis for the use of endophyte in wild rice in the future.

## RESULTS

### Original sequencing data and α-diversity

The original metagenome sequencing data are shown in Table S1. The Q20 values of all the samples were greater than 98.67%, and the Q30 values were all greater than 95.32%. The proportion of low-quality reads in the samples was 0 (29). The α-diversity analysis revealed that the Shannon (Fig. 1A) and Chao1 (Fig. 1B) indices of *O. granulata* were significantly higher than those of cultivated rice. The statistical significance was assessed using a two-tailed Student's *t*-test ($P < 0.01$). No significant difference was observed between the two wild rice species. Compared with cultivated rice, *O. officinalis* had higher values for both indices, but the differences were not statistically significant ($P > 0.05$).

### Microbial community structure

Species annotations were performed using the sequenced gene set. Among the nine samples, four domains were annotated: 14 phyla, 23 classes, 39 orders, 58 families, 76 genera, and 102 species. All sample taxonomic phylogenetic trees are shown in Fig. S1. As shown in Table S2, at the boundary classification level, the leaf endophytes were mainly bacteria but also included archaea, viruses, and fungi. The relative abundances of archaea and viruses in the leaves of *O. meyeriana* were higher than those in the other types of rice. The statistical significance was assessed using a two-tailed Student's *t*-test ($P < 0.05$). *Pseudomonadota*, *Bacillota*, *Actinomycetota*, and *Artverviricota* accounted for the largest proportions at the phylum level (Fig. 1C). Among the three groups of samples, these four phyla accounted for 97.55–98.67% of the total abundance. Among them, *Pseudomonadota* showed the highest proportion, accounting for over 60% in all three groups of samples. Notably, its proportion was the highest in cultivated rice (84.83%). The most abundant genera were *Dechloromona*, *Salmonella*, *Klebsiella*, *Listeria*, and *Clostridioides* (Fig. 1D), and the predominant endophytes at the species level were *Dechloromonas* sp. H13, *Salmonella enterica*, *Klebsiella pneumoniae*, *Listeria welshimeri*, and *Clostridioides difficile* (Fig. 1E). The proportions of microorganisms of the different genera or species mentioned above were not the same among the different groups. For example, the relative abundance of *S. enterica* in *O. officinalis* was considerably lower than that in cultivated rice. The statistical significance was assessed using a two-tailed Student's *t*-test ($P < 0.01$).

### Microbial community structure of leaf endophytes in wild and cultivated rice

To explore the differences in endophytes among different *Oryza* species, the relative abundances of microflora in the three groups of rice were compared and analyzed using linear discriminant analysis effect size (LEfSe) to determine the microflora that changed significantly ($P < 0.05$). The linear discriminant analysis (LDA) score was used to estimate the effect of the abundance of each species (Fig. 2A). LEfSe showed that the abundance of *Gammaproteobacteria* was significantly higher in cultivated rice than in the two types of wild rice (LDA score, 5.51). The three classes, *Bacilli* (LDA score, 4.83), *Betaproteobacteria* (LDA score, 4.58), and *Clostridia* (LDA score, 4.53), were dominant in *O. officinalis*. *Ligilactobacillus salivarius* (LDA score, 4.46) was dominant in *O. meyeriana*. When compared with the other two groups of samples, *Archaea* was dominant in *O. meyeriana* (LDA score, 4.01). The differences between the two groups of wild rice were analyzed separately (Fig. 2B); the dominant species in *O. meyeriana* included *L. salivarius* (LDA score, 4.44) and *Bradyrhizobium* sp. MOS002 (LDA score, 4.38). *Acinetobacter baumannii* (LDA score, 4.41) and *Dechloromonas* sp. H13 (LDA score, 4.41) were dominant in *O. officinalis*.

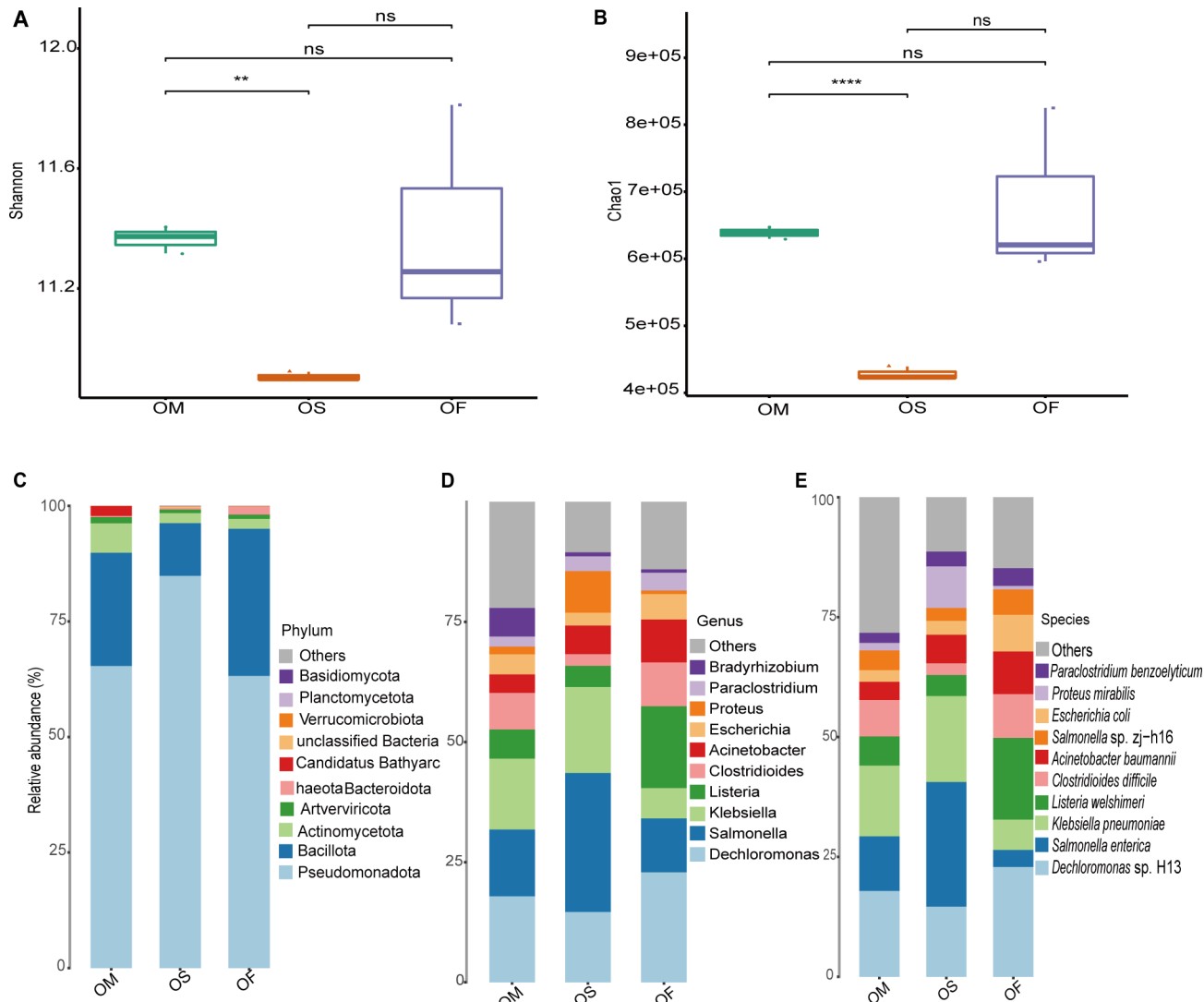

**FIG 1** Shannon (A) and Chao1 (B) indices for each rice type. Values are means ± SD from three biological replicates. Statistical significance was assessed using a two-tailed Student's *t*-test (**, $P < 0.01$). Taxonomic profiles of microbial communities in wild and cultivated rice leaf endophytes. The relative abundances of the endophytes dominant phyla (C), genera (D), and species (E) in each rice type. OS, cultivated rice; OF, *O. officinalis*; OM, *O. meyeriana*.

Random forest is an ensemble-learning model made up of numerous decision trees. During construction, it randomly samples data and features and finally integrates the results of all trees, thereby reducing the risk of overfitting and efficiently handling complex data (30). Based on the random forest model, the importance of the top 20 genera affecting the difference in distribution between two species of wild rice and cultivated rice was predicted separately. At the genus level, the top four genera with the highest contribution to the interspecific differences between *O. meyeriana* and cultivated rice were *Streptococcus*, *Chitinimonas*, *Sphingobium*, and *Ligilactobacillus* (Fig. 2C). In contrast, the top four genera contributing to the greatest differentiation between *O. officinalis* and cultivated rice were identified as *Proteus*, *Mycobacteroides*, *Symbiopectobacterium*, and *Salmonella* (Fig. 2D).

Principal coordinate analysis (PCoA) based on the Bray-Curtis distance showed that the composition of leaf endophyte communities in the different types of rice was significantly different according to the rice genotype ($R^2 = 0.96$, $P = 0.006$) (Fig. S2A). The hierarchical cluster analysis at the microbial genus level showed that *O. meyeriana* and *O. officinalis* clustered together, whereas cultivated rice was a separate branch. This indicated that among the three types of rice, the leaf endophytic community structures

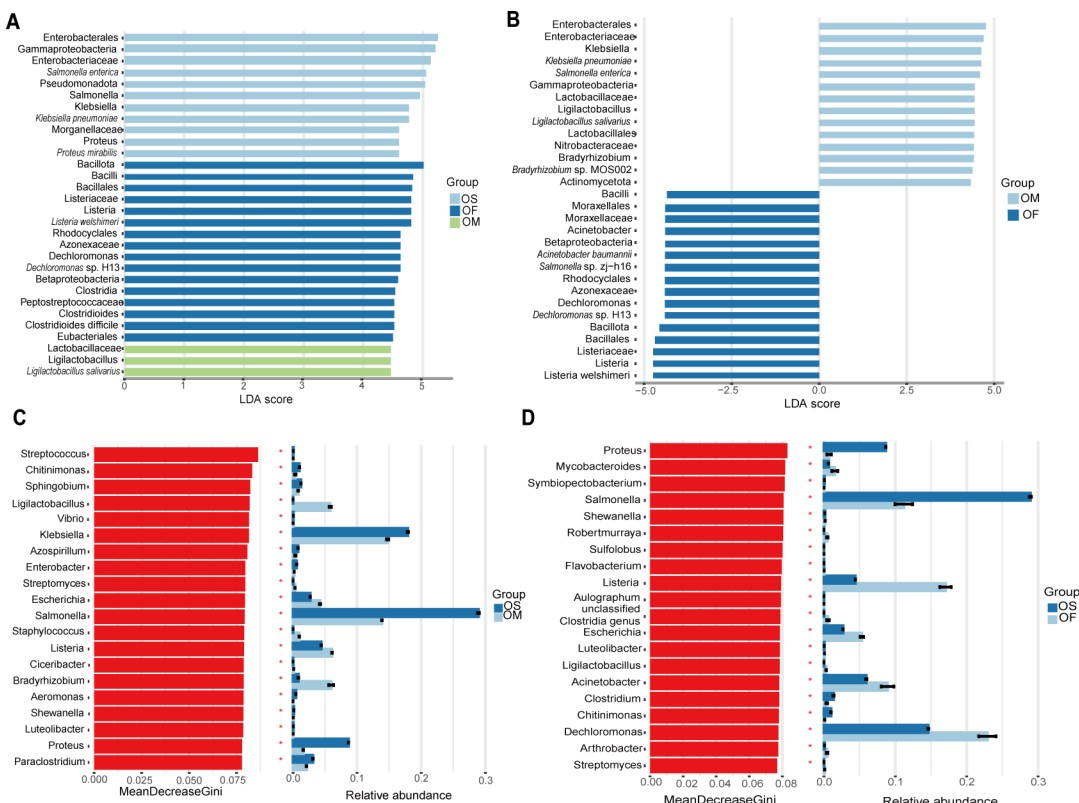

**FIG 2** LEfSe analysis of different types of rice (*P* < 0.05). The microbial compositions among three types of rice (A) and between two wild rice species (B) were analyzed through LEfSe. Based on the random forest classification model and its relative abundance, the species affecting the differential distribution between *O. meyeriana* and cultivated rice (C) and between *O. officinalis* and cultivated rice (D) were identified.

of *O. officinalis* and *O. meyeriana* exhibited high similarity but differed from that of cultivated rice (Fig. S2B).

## Identification of endophyte core microorganisms in rice leaves

Through species annotation and Venn analysis, we found 48 species of core microorganisms in the three groups of samples (Fig. S3). The average relative abundance of these core microorganisms in the three groups was 98.12%. *Dechloromonas* sp. H13 was the most abundant species (18.47%), followed by *S. enterica* (13.64%), *K. pneumoniae* (13.00%), *L. welshimeri* (9.17%), and *C. difficile* (6.38%). *Pseudomonadota* and *Bacillota* were the two most abundant phyla among the endophyte core microorganisms in rice leaves. Although the abovementioned core endophytes were distributed in the three groups of samples, the distribution of these microorganisms in different types of rice was not consistent. The relative abundances of *A. baumannii*, *Dechloromonas* sp. H13, and *L. welshimeri* in *O. officinalis* were significantly higher than those in *O. meyeriana* and cultivated rice. The relative abundance of *S. enterica* in *O. meyeriana* was significantly higher than that in *O. officinalis* and cultivated rice. The statistical significance of all relative abundances was evaluated using a two-tailed Student's *t*-test (*P* < 0.01) (Table S3).

## Co-occurrence network analysis of endophytic bacterial communities in leaves

To explore the relationships between endophytes in the leaves of different types of rice, a co-occurrence network analysis of their microbial communities was performed (Fig. 3A through C). The results showed 46, 43, and 43 nodes and 350, 401, and 288 edges in the microbial collinear networks of *O. officinalis*, *O. meyeriana*, and cultivated rice,

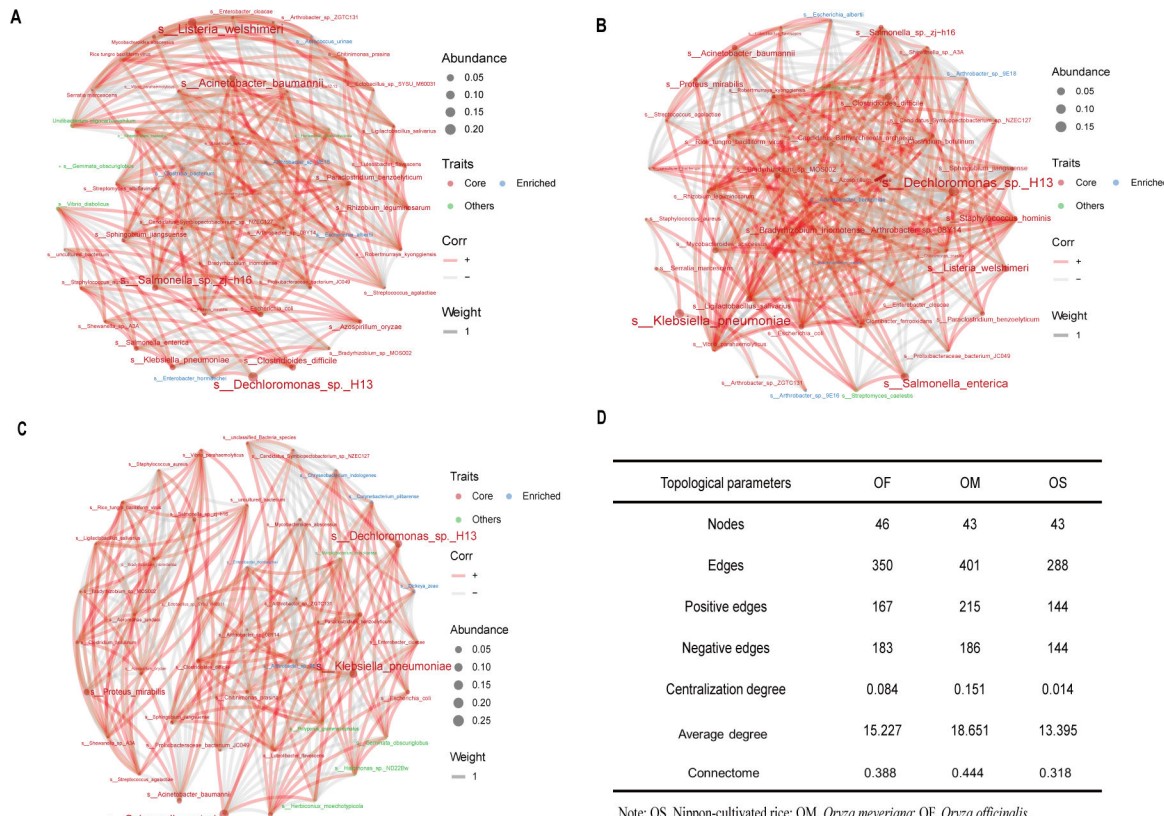

**FIG 3** Co-occurrence network of endophytes in different types of rice. Co-occurrence network of endophytes in *O. officinalis* (A), *O. meyeriana* (B), and cultivated rice leaves (C). Positive and negative relationships are illustrated in red and green, respectively. Topological parameters of co-occurrence networks in different types of rice (D).

| Topological parameters | OF | OM | OS |
|---|---|---|---|
| Nodes | 46 | 43 | 43 |
| Edges | 350 | 401 | 288 |
| Positive edges | 167 | 215 | 144 |
| Negative edges | 183 | 186 | 144 |
| Centralization degree | 0.084 | 0.151 | 0.014 |
| Average degree | 15.227 | 18.651 | 13.395 |
| Connectome | 0.388 | 0.444 | 0.318 |

Note: OS, Nippon-cultivated rice; OM, *Oryza meyeriana*; OF, *Oryza officinalis*.

respectively. The topological parameters of the co-occurring networks of different types of rice calculated using Gephi are listed in Fig. 3D. The network topological characteristics of the three types of rice were different. The number of edges and the average degree of *O. meyeriana* were higher than those of *O. officinalis* and cultivated rice. This suggests that the endophytes in the two types of wild rice were more closely associated than with those in cultivated rice and that the network structure was more complex. Except for *O. officinalis*, positive associations were higher than negative associations in the remaining networks. The analysis also revealed that the leaf endophytic microbial community in cultivated rice had the highest degree of modularity (0.661).

## Predicted functional profiles of leaf endophytes

The function of the non-redundant gene set obtained from the three groups of samples was predicted using the Kyoto Encyclopedia of Genes and Genomes (KEGG) database. At the KEGG pathway level 1, the average read proportion of metabolic classes in nine samples was 76.10%. The reads of genetic and environmental information processing, cellular processes, human diseases, and organismal systems categories accounted for 0.95, 7.80, 2.72, 7.53, and 4.90% of the total functional annotation reads, respectively. At the KEGG pathway level 2, the most abundant functional category in the metabolic category was the global and overview map, followed by energy, carbohydrate, and amino acid metabolisms. In the functional classification of genetic information processing, the abundance of folding, sorting, and degradation was the highest, followed by transcription, translation, replication, and repair. Among the functional categories of environmental information processing, the relative abundance followed the order of signal transduction > membrane transport. The functional classification

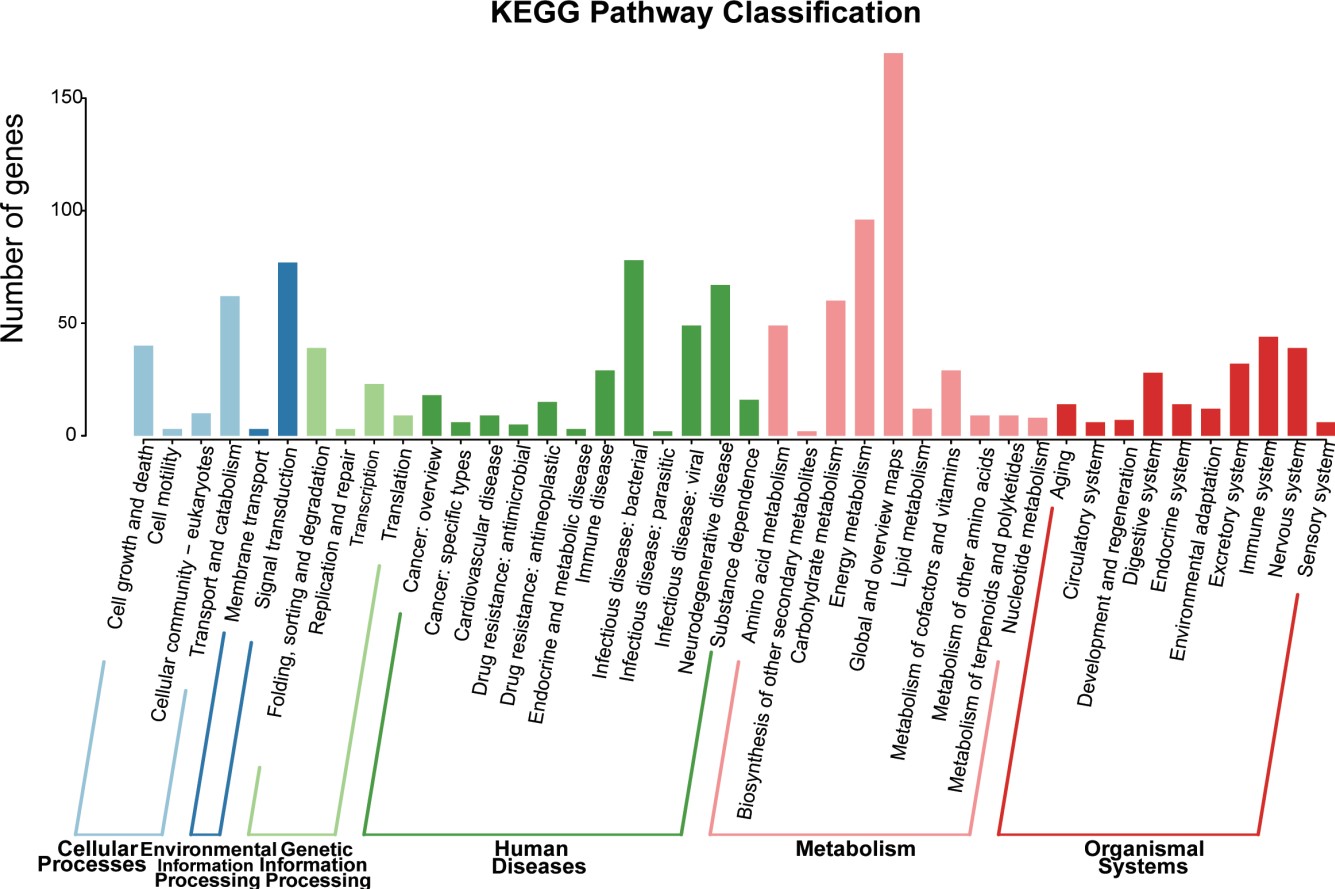

**FIG 4** Classification of the KEGG pathways. The horizontal axis is the name of the metabolic pathway involved, and the vertical axis is the number of genes annotated to that pathway.

that belongs to the category of human diseases with a higher abundance was infectious disease: bacterial and neurodegenerative diseases. This phenomenon might be primarily attributed to the human-centered construction of the KEGG database, leading to the non-specific mapping of bacterial genes through sequence homology. In the functional classification of biological systems, the nervous and immune systems were most abundant. In the functional classification of the cellular processes, transport and catabolism were most abundant (Fig. 4).

## Comparison and difference analysis of microbial functions of the three types of *Oryza*

The abundance of each KEGG functional level in different groups of samples was determined, and the functional diversity of the endophytic microbial communities in the different type of rice was analyzed. According to the results, differences were observed among the pathway, module, and orthology diversity of the microbial communities in the different types of rice. The diversity order of the pathway, module, and orthology of the microbial community in these three groups of samples indicated that the diversity in *O. officinalis* was greater than that in *O. meyeriana* and cultivated rice. The analysis of the KEGG orthology groups (KOs) of the three groups of samples and the comparisons using a Venn diagram (Fig. S4A) showed that the number of specific KOs was the highest in *O. meyeriana* (9), followed by that in *O. officinalis* (6).

The heatmap was constructed for the three groups of samples, and the functional differences of microorganisms among the different samples were analyzed. Forty-two functional categories annotated at the pathway level 2 based on the KEGG classification

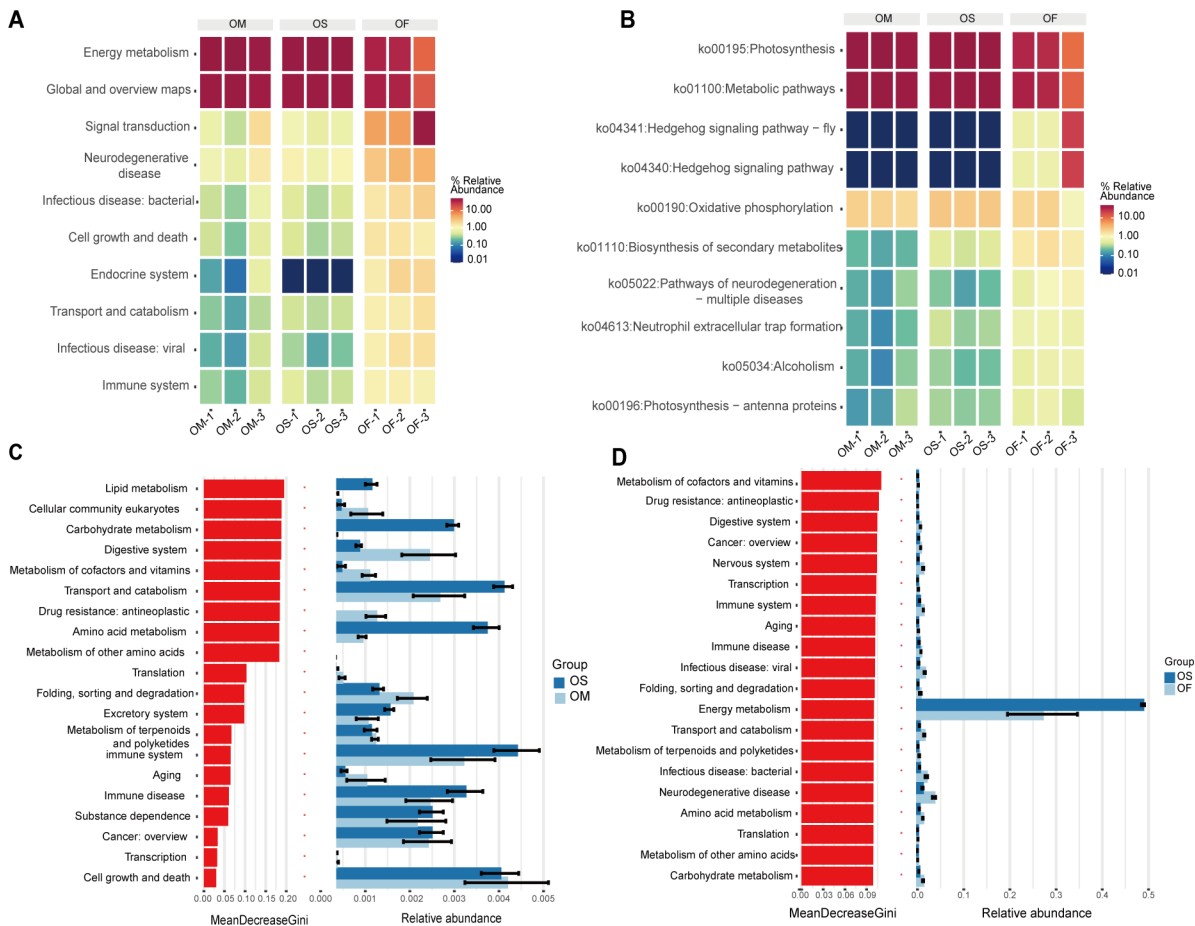

**FIG 5** KEGG orthology (KO) group comparisons. The heatmap shows the top 20 relative abundances of the dominant pathways at pathway levels 2 (A) and 3 (B) in each group. Based on the random forest classification model and its relative abundance, the pathways at level 2 affecting the differential distribution between *O. meyeriana* and cultivated rice (C) and between *O. officinalis* and cultivated rice (D) were identified.

system were analyzed and compared (Fig. 5A). The relative abundances of metabolic categories were dominant, including energy metabolism and global and overview maps. At the KEGG pathway level 3, the dominant pathways of the leaf endophyte microbial community were photosynthesis, metabolic pathways, the Hedgehog signaling pathway-fly, oxidative phosphorylation, and biosynthesis of secondary metabolites (Fig. 5B). According to the heatmap, it can be observed that the relative abundances of photosynthesis (PATH: ko00195) and metabolic pathways (PATH: ko01100) are similar in *O. meyeriana* and cultivated rice and higher than those in *O. officinalis*. Most of the pathways at levels 2 and 3 showed that the relative abundance in *O. officinalis* was higher than those in the other two types of rice.

Based on the KEGG classification system and random forest analysis, at the pathway level 2, the functions that differed between *O. meyeriana* and cultivated rice included lipid metabolism, eukaryotic cellular community, and carbohydrate metabolism (Fig. 5C). Nine functions, including transport, catabolism, and amino acid metabolism, were more abundant in cultivated rice. A higher relative abundance of the functions, such as the digestive system, drug resistance, and antineoplastics, was noted in *O. meyeriana*. Pathways, including the metabolism of cofactors and vitamins, drug resistance (antineoplastic), and digestive system, had a higher contribution in the *O. officinalis* and cultivated rice groups (Fig. 5D). In addition, the relative abundances of the remaining 19 pathways were higher in medicinal wild rice than in cultivated rice, except for energy metabolism.

In the hierarchical clustering analysis of KEGG based on the Bray-Curtis distance algorithm (Fig. S4B), *O. officinalis* and cultivated rice clustered together, whereas *O. meyeriana* was in another branch. The principal component analysis reflects the differences and distances between samples. PCoA showed that the metabolic function of the endophytic microbial community differed among the groups of rice (Fig. S4C). According to the results of the KEGG function annotation, it was extracted into two principal components at level 3, and the contribution rate of PCo1 was 75.7%. The contribution of $PCo_2$ was 19.4% ($R^2 = 0.89$, $P = 0.007$).

## DISCUSSION

In this study, through PCoA based on the Bray-Curtis coefficient, LEfSe, and random forest analysis-based group comparison, differences in endophytic diversity and relative abundance of different microorganisms due to host genotype and species changes were observed. These results showed that the genotype of wild rice had a marked impact on the endophyte community structure. Although host plant species are a determinant of the phyllosphere community structure, air serves as a key transmission route for phyllospheric bacteria (31), and neighboring plants or plant debris can also drive differences in the composition of endophytic microbial communities (32). This could potentially explain the differences observed between this study and previous research (33).

Plants are colonized and interact with diverse microorganisms that affect their physiology, growth, and health. Holobionts formed by plants and their related core microorganisms have important effects on the overall stability and adaptability of plants (34, 35). The core microbiome has been studied in various plants, such as *Salvia miltiorrhiza*, *Saccharum officinarum*, and *Arabidopsis thaliana* (36–39). The core microbiome identified here includes beneficial microorganisms with known functions in plant growth promotion and stress resistance. For instance, *Azospirillum*, a well-studied plant growth-promoting bacterium can fix nitrogen from the air and produce plant hormones, such as auxins, cytokinins, and gibberellins (40). *Prolixibacteraceae*, mainly involved in the nitrogen cycle, have the ability to reduce nitrate to nitrite in hypoxic conditions (41). *Acinetobacter* can improve plant root absorption of nitrogen and phosphorus to promote growth (42); it can also repair pesticide residues in the soil (43) and enhance plant resistance and growth under heavy metal stress (44). *Streptomyces* exhibits the highest biocontrol activity compared to other *Actinomycetes* species and inhibits various pathogens (45, 46). Furthermore, current core microorganism studies focus on the genus level and above. For example, Hamonts et al. (38) reported that key microorganisms in sugarcane leaves are *Cladosporium*, *Periconia*, *Nigrospora*, and *Bullera*. Although these studies identify the core microbiome at a certain taxonomic level, they provide only limited assistance for the design of artificially assembled microbiota. In the present study, the core microbiome was identified at the species level, and most of these species have multiple beneficial effects on rice growth. Therefore, the structure of the beneficial core microbiome can be managed in the future for high and stable rice yields.

Core microbiome analyses detected pathogens, such as *Salmonella* and *Listeria*. Surface sterilization methods aim to reduce contamination, but these taxa likely reflect endophytic colonization as some pathogens can enter plant tissues via stomata or wounds (47). A key limitation is that molecular methods cannot distinguish viable cells from non-viable ones; therefore, the detection does not confirm active infection (48). Ecologically, these bacteria may act as transient commensals in plants, using nutrients without causing symptoms, as it has been observed in crop endophytes (49). However, their presence suggests functional plasticity as the stress (e.g., wounding and nutrient stress) may trigger pathogenicity. Future studies using multiomics and inoculation assays are needed to clarify their roles.

Co-occurrence network analysis showed that the correlation between the endophytic communities of *O. meyeriana* and cultivated rice is mainly positive, indicating that most microbial members cooperate, including having nutritional relationships, and

may share an ecological niche based on their preferences and functional characteristics (50). However, the endophytic communities of *O. officinalis* leaves have a higher negative correlation, suggesting that the ecological competition may increase microbial community stability by reducing the destabilizing effect of cooperation (51). In addition, the wild rice networks have a higher connectome and average degree than those of cultivated rice. These results imply that wild rice has a better structure and more stable community than cultivated rice perhaps due to less human interference or host genotype differences (52). This result is supported by previous studies showing that more complex microbial coexistence patterns are strongly associated with higher functional community characteristics (53).

The relative abundance (2.12%) of unclassified *Candidatus Bathyarchaeota* in *O. meyeriana* was considerably higher than in the other rice samples, whereas its relative abundance in cultivated and medicinal wild rice was nearly 0. *Candidatus Bathyarchaeia* is widely distributed in various ecosystems such as high-salt (54), marine (55), and freshwater sediments (56), and is especially abundant (up to 100% of total archaeal abundance) in marine sediments (57). Its ubiquity and predominance in natural anaerobic microbial communities may result from its ability to metabolize many organic substrates, such as detrital proteins, aromatic compounds, lignin, and extracellular carbohydrates (58–60). The sclerenchyma of the main and lateral veins of *O. meyeriana* is more developed and exhibits a high degree of lignification with a large amount of lignin present (61). This lignin content serves as a source of raw material for the metabolic growth of *Candidatus Bathyarchaeota*, potentially contributing to its higher relative abundance in *O. meyeriana* in this study. The relative abundance of *Bradyrhizobium* was the highest (5.94%) in *O. meyeriana* among the three samples. As an important member of the soil bacteria, collectively known as rhizobia, *Bradyrhizobium* has multiple biochemical functions, such as biological nitrogen fixation and carbon fixation, and is a common taxon in the soil (62, 63). However, recent studies have shown that although some non-symbiotic *Bradyrhizobium* isolates lacking nodule gene clusters are unable to nodulate legumes, they are commonly found in the soil, as their genomes lack functional genes related to nodulation and nitrogen fixation (64). Therefore, the reason for its accumulation in the leaves of *O. meyeriana* is unknown, and its function in nitrogen fixation needs further identification.

Endophytes are abundant sources of naturally active metabolites, and many secondary metabolites with antibacterial, antifungal, insecticidal, antioxidant, and anticancer activities have been identified. They possess unique genetic and metabolic diversities and have potential for medical and pharmaceutical applications (65, 66). Analysis of the KEGG functional annotation results showed that the relative abundance of biosynthesis of secondary metabolite pathways annotated in *O. officinalis* was five times higher than that in *O. meyeriana* and two times higher than that in cultivated rice. These results indicate that endophytes from wild rice have good research potential and are potential resources for antibiotic development. Subsequently, related strains, such as *Acinetobacter* and *Streptomyces*, can be obtained by traditional isolation and culture methods. The *O. officinalis* growth was the highest among the 22 species of rice and more than 20 times that of commonly cultivated rice. The results of this study are limited in their ability to fully elucidate this phenomenon. Specifically, it was observed that a higher proportion of endophytes with nitrogen-fixing and phosphorus-solubilizing effects were present among the medicinal wild rice endophytes, including *Azospirillum*, *Bradyrhizobium*, and *Rhizobium*.

We acknowledge the limitation regarding the small sample size ($n$ = 3 per group), which may indeed restrict the statistical value of our analyses. In addition, the environment can exert a substantial effect on the community structure of endophytic bacteria (66), which may lead to some limitations in the results presented in this study. Due to the current paucity of research on leaf endophytic bacteria in wild rice, it is difficult to validate the findings of this study. Nevertheless, this study furnishes exploratory insights and data-driven support for subsequent analyses concerning the community

structure of leaf endophytic bacteria in wild rice. In this study, microbial functions were primarily predicted through KEGG annotations. Consequently, the inferred pathway activities lack validation from metatranscriptomic and metabolomic analyses. To address this, subsequent experiments incorporating these approaches are essential to assess the accuracy of the predicted functions. Such validation will bridge the gap between predictive annotations and actual biological processes, thereby enhancing the mechanistic understanding of the observed phenomena.

In conclusion, the community composition of endophytic bacteria in the leaves of wild rice was significantly different from that in the leaves of cultivated rice, and differences were observed between the two types of wild rice. The community composition of wild rice was more complex and diverse than that of cultivated rice. Our study verified the core microbiome and listed potentially beneficial endophytes and observed the general functions of endophytes. The preliminary identification of the composition of the endophyte community in wild rice leaves is a crucial step toward extracting beneficial endophytes from wild rice to promote the sustainable development of rice yield in the future.

## MATERIALS AND METHODS

### Sample collection and processing

Wild rice samples were collected from Menghai County (21°58′N, 100°28′E) and Mengla County (21°27′32″N, 101°33′50″E) in Xishuangbanna Dai Autonomous Prefecture, Yunnan Province and preserved in the greenhouse of the Yunnan Academy of Agricultural Sciences in Kunming (24°82′76″N, 102°84′52″E), China. In September 2023, two or three leaves of two types of healthy wild rice were sampled, and the experiment was repeated three times. Leaf samples of Nippon-cultivated rice were used as controls. Samples were sequentially washed in 70% ethanol for 2–5 min in 2.5% sodium hypochlorite to eliminate surface microbial contaminants. Subsequently, samples were rinsed one to three times with sterile water, and residual moisture on the tissue surface was absorbed using sterile filter paper. The final rinse water was plated onto Luria-Bertani agar plates to verify the efficacy of surface sterilization. The samples were rapidly frozen in liquid nitrogen and stored at –80°C for subsequent metagenomic sequencing. Subsequent DNA extraction and sequencing were conducted by Sangon Biotech (Shanghai) Co., Ltd.

### DNA extraction

Total DNA was extracted from leaf tissues using the EZNA Soil DNA Kit (Omega Bio-Tek, M5635-02) following the manufacturer's protocol. DNA concentration and purity were assessed using a Qubit 4.0 fluorometer (Thermo Fisher Scientific) and a NanoDrop One spectrophotometer (Thermo Fisher Scientific), respectively, to ensure high-quality input for downstream analyses.

### Library preparation

Sequencing libraries were prepared with 500 ng DNA using the Hieff NGS MaxUp II Kit (Yeasen, 12200ES96). DNA was fragmented to approximately 500 bp via Covaris M220 ultrasonication, followed by end-repair, adaptor ligation (Illumina dual-index adapters), and PCR amplification (12 cycles, 2× Super CanaceII High-Fidelity Mix). Libraries were size-selected (300-600 bp) with magnetic beads (Yeasen, 12601ES56) and quantified using Qubit 4.0 before paired-end sequencing (2 × 150 bp) on a NovaSeq 6000 platform (Illumina).

## Data assessment and quality control

The sequenced data were quality-assessed using Fastp (version 0.36). The raw reads were processed as follows: (i) adapter sequences were removed; (ii) low-quality bases (Q < 20) from the 3′ to 5′ end of reads were eliminated by a sliding window approach with a window size of 4 bp to filter out tail-end bases with values less than 20; (iii) inconsistent bases within overlaps of paired reads were corrected; and (iv) reads shorter than 35 nt and their paired reads were discarded. The resulting high-quality data (Q30 ≥ 90%) were used for subsequent analysis.

## Metagenome assembly

Initially, Megahit (version 1.2.9, k-mer range 21–129) was used for multi-sample mixed splicing to obtain preliminary contig sequences. Subsequently, bowtie2 (version 2.1.0) was used to align the clean reads with the spliced results, and the unaligned reads were isolated and further subjected to splicing using SPAdes (version 3.13) to generate contigs of low abundance.

## Gene prediction and non-redundant gene set construction

The Prodigal software (version 2.60) was used to predict open reading frames of splicing outcomes and select genes with lengths of at least 100 bp for translation into amino acid sequences. After gene prediction for each sample, CD-HIT (version 2.60) was used to remove redundancy and obtain a unique set of genes. Then, Salmon (version 1.5.0) was used to create a specific index of non-redundant gene sets using a dual-phase algorithm and a bias model construction method to accurately quantify gene abundance in each sample based on gene length information.

## Species and functional annotations

The gene set was compared against KEGG (2023 release) and other databases via Diamond (version 0.8.20) to obtain species and functional annotation information for the genes. The screening criteria encompassed $E$-value < 1e−5 and a score > 60. Species and functional abundances were ascertained based on the gene set abundance and annotation information, followed by multilateral statistical analyses, such as species and functional composition analysis, species and functional difference analysis, and sample comparison analysis.

## Statistical analysis

All statistical analyses were performed using R software (version 4.0.3). The β-diversity evaluates variations among samples in the microbiome and is combined with dimensionality reduction technique, PCoA, for visualizations. Analyses were performed with the R vegan package (version 2.5-6) and shown as scatterplots. Metastats (Wilcoxon, FDR < 0.05) examines microorganisms with significant differences between two groups. LEfSe (version 1.1.0) analyzes differential microorganisms and functions among multiple groups. Co-occurrence networks included species with relative abundances > 0.05% constructed via Spearman correlations ($|r| > 0.6$, $P < 0.01$) and analyzed for modularity (Louvain algorithm) and topological properties (Gephi v0.9.2).

## ACKNOWLEDGMENTS

This study was supported by Yunnan Seed Laboratory (202205AR070001-01), the National Key R&D Program of China (2021YFD1200100), Science and Technology Personnel and Platform Plan Field Scientific Observation Research Station Construction Project (202205AM340037), and the Open Research Fund of State Key Laboratory of Crop Gene Exploration and Utilization in Southwest China (SKL-KF-202325).

L.L.: investigation, data curation, and writing—original draft; X.L.—review and editing; Z.X.—review and editing; J.L.: data duration; L.C.: data curation; Q.Z.: investigation; H.J.:

investigation; Z.C.: conceptualization, resources, funding acquisition, writing—review and editing, and supervision; and S.X.: conceptualization, methodology, and writing—review and editing.

## AUTHOR AFFILIATIONS

[1]Biotechnology and Germplasm Resources Institute, Yunnan Academy of Agricultural Sciences, Kunming, China

[2]Ministry of Agriculture Key Laboratory of Molecular Biology of Crop Pathogens and Insects, Institute of Biotechnology, Zhejiang University, Hangzhou, China

## AUTHOR ORCIDs

Lingyun Lei  http://orcid.org/0009-0005-0748-0398
Zaiquan Cheng  http://orcid.org/0000-0002-6897-1255
Suqin Xiao  http://orcid.org/0000-0002-7130-973X

## FUNDING

| Funder | Grant(s) | Author(s) |
| --- | --- | --- |
| Yunnan seed Laboratory | 202205AR070001-01 | Li Liu |
| | | Zaiquan Cheng |
| Science and technology personnel | 202205AM340037 | Li Liu |
| | | Zaiquan Cheng |
| National Key Research and Development Program of China | 2021YFD1200100 | Qiaofang Zhong |
| | | Zaiquan Cheng |
| Open Research Fund of State Key Laboratory of Bioelectronics | SKL-KF-202325 | Jinlu Li |

## AUTHOR CONTRIBUTIONS

Lingyun Lei, Writing – original draft | Xingyi Li, Resources | Zixuan Xiong, Validation | Jinlu Li, Validation | Li Liu, Project administration | Ling Chen, Software | Qiaofang Zhong, Project administration | Hui Jiang, Methodology | Zaiquan Cheng, Writing – review and editing | Suqin Xiao, Writing – review and editing

## DATA AVAILABILITY

The original sequences have been deposited in the NCBI Sequence Read Archive (SRA) under BioProject ID number PRJNA1119411.

## ADDITIONAL FILES

The following material is available online.

### Supplemental Material

**Supplemental material (Spectrum02978-24-S0001.docx).** Supplemental figures and tables.

### Open Peer Review

**PEER REVIEW HISTORY (review-history.pdf).** An accounting of the reviewer comments and feedback.

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
