## [Reviewer comments · Microbiology Spectrum]

Microbiology Spectrum

Distinctive structure of endophytic microbial communities in two species of wild rice and cultivated rice

lingyun lei, Xingyi Li, Zixuan Xiong, Jinlu Li, Li Liu, Ling Chen, Qiaofang Zhong, Hui Jiang, Suqin Xiao, and Zaiquan Cheng

Corresponding Author(s): Zaiquan Cheng,

Review Timeline:

Submission Date:	November 19, 2024
Editorial Decision:	April 8, 2025
Revision Received:	June 10, 2025
Accepted:	June 17, 2025

Editor: Lindsey Burbank

Reviewer(s): Disclosure of reviewer identity is with reference to reviewer comments included in decision letter(s). The following individuals involved in review of your submission have agreed to reveal their identity: Malick Bill (Reviewer #1); Jacob A. Heil (Reviewer #3)

Transaction Report:

DOI: <https://doi.org/10.1128/spectrum.02978-24>

Re: Spectrum02978-24 (Distinctive structure of endophytic microbial communities in two species of wild rice and cultivated rice)

Dear Miss lei ling yun:

Thank you for the privilege of reviewing your work. First of all, I would like to apologize for the length of time your manuscript was in review. I took over from the editor who was initially assigned because they unexpectedly were not able to complete the assignment. Please consider the comments of the reviewers as outline below and in the attached documents. The most substantial criticisms that will need to be addressed are related to statistical analysis and discussing the limitations of the experimental design. In addition, the reviewers were very thorough in assessing your manuscript, and careful attention should be paid to all of the other comments such as checking and adding relevant references.

Below you will find my comments, instructions from the Spectrum editorial office, and the reviewer comments.

You will receive reminders to submit your revised manuscript within 60 days; but if you cannot complete the modification within this time period, please contact me and I can extend that deadline. If you do not wish to modify the manuscript and prefer to submit it to another journal, notify me so that the manuscript may be formally withdrawn from consideration by Spectrum.

Revision Guidelines

Sincerely,
Lindsey Burbank
Editor
Microbiology Spectrum

Reviewer #1 (Comments for the Author):

There are some sections of the results section where the authors claim differences between the groups of plants but without providing statistical evidence of those differences as indicated in Line ii5-116 and Line 177.

Reviewer #2 (Comments for the Author):

This study investigates the endophytic microbial communities in leaves of two wild rice species (*Oryza officinalis* and *O. meyeriana*) and cultivated rice (*O. sativa*) using metagenomic sequencing. Key findings include higher microbial diversity and network complexity in wild rice, identification of core microbiomes (e.g., *Dechloromonas*, *Salmonella*, *Klebsiella*), and enriched metabolic pathways in wild species. The authors suggest wild rice endophytes may offer beneficial traits for improving crop resilience and sustainability. Although the findings of the research are generally interesting, I have following concerns that should be addressed before considering for publication:

1. The small Sample size (n=3 per group) limits statistical power. Please acknowledge this limitation and consider supplemental data (e.g., public datasets) to validate trends.
2. Another issue is reliance on predictive KEGG annotations without metatranscriptomic/metabolomic validation. I suggest the authors to highlight this as a caveat and propose future experiments to confirm pathway activity.
3. What is the meaning of the presence of pathogens (*Salmonella*, *Listeria*) in core microbiomes without addressing contamination risks or host interactions? Discuss sterilization efficacy and potential plant-pathogen dynamics (e.g., commensalism vs. pathogenesis).
4. Why do you focus on leaves that overlook root-associated microbes critical for nutrient uptake?
5. Claiming about *Bradyrhizobium*'s role in leaves lack experimental support. Explain.
6. Simplify visuals (e.g., merge panels, highlight key taxa) and clarify axis labels/legends.
7. Although generally clear but requires proofreading for minor grammatical errors (e.g., inconsistent hyphens, tense shifts).
8. Specify p-values for PCoA/perMANOVA results and clarify significance thresholds in LEfSe.
9. Overall, this manuscript provides valuable insights into the microbiome divergence between wild and cultivated rice, supported by robust metagenomic data. While methodological limitations exist, the findings lay a foundation for harnessing wild rice microbiomes to enhance crop resilience. With revisions addressing the above points, this study merits publication in *Microbiology Spectrum*.

Reviewer #3 (Comments for the Author):

Reviewer summary:

The authors of this study aim to build a base of knowledge about the microbiomes of leaves from different rice species including two wild and one domestic. They approach this characterization with the goals of describing species composition, comparing microbiomes across species, characterizing and comparing the functional profiles of these communities, and building a molecular network interaction model for the rice and their microbiomes. The subject matter is timely and important. Descriptive studies of plant microbiomes are important bases of knowledge for applied experiments and the data presented here is valuable for that reason. However, the study design is not sufficient to support many of the conclusions of the authors. For example, they do not seem to account for site analyses when that may be as or more important than host species in determining leaf microbiome structure. There is interesting data here, but the current manuscript needs extensive revision.

Comments by line:

Line 26-28 Inconsistent terminology. First "abundances" is used and then "relative abundance". If you are referring to relative abundance please always qualify it as relative abundance.

Line 49-51 Fundamentally false. There are many endophytic pathogens. The source does not support this statement that endophytes "do not cause negative effects, injuries, or symptoms in their hosts".

Line 51-52 This sentence is overly broad and the source does not support the statement that "most endophytes assist in plant growth". First, endophytic microbiomes vary by plant species and many other factors. Second, demonstrating that the majority of endophytes in all plants contribute to "growth" (also a broad term) would require an incredible level of proof.

Line 57-59 Needs source. The statement "endophytes are the main factors that affect plant growth and development" is false.

Line 59-63 Not a comprehensive list. Reword to say "some confirmed functions"

Line 66-67 Needs source.

Line 79-80 The source is about beets but the author is making a statement about rice. How is valuable defined in this context?

Line 80-82 Source does not support sentence.

Line 86 Questionable and simplistic as a universal statement. Sentence should state that this is the finding of one study.

Line 112-113 Source needed

Line 115-117 No description of how significance was determined. Needs test statistics and p values. All of this is also missing from the figure legend.

The entire results section needs more detail. Results are often reported qualitatively with no quantification. Many sentences refer to figures but do not have the figure referenced (e.g., lines 124-125 and 127).

Line 150 First reference to a random forest model. What is the model structure?

Lines 257-259 Needs source

Lines 261-263 Unsubstantiated. Data in this study were not sufficient to demonstrate that any associations between microbes and their host species pre-date rice domestication.

Lines 278-281 Confusing. I am not sure what this sentence is trying to say.

Lines 282-284 Needs source.

Lines 329-332 Sentences need sources.

The materials and methods section needs to be more detailed. Describe all models in detail. The metadata do not seem sufficient to draw many of the conclusions. For example, many of the differences in this study may be attributable to site instead of plant species, however, study design and statistical methods do not seem to account for the effect of site. Other variables to consider incorporating include time of year, weather, and age of plants.

The authors of the manuscript titled “Distinctive structure of endophytic microbial communities in two species of wild rice and cultivated rice” compared the endophytic communities of two wild rice and cultivated rice using a metagenomics approach and further provided theoretical basis for future utilization of endophyte in wild rice. The work is interesting and clearly motivated. However, I have several concerns as indicated below.

Line 75: Please write “*O. meyeriana*” in full at the beginning of the sentence.

Line 105: A full stop is missing at the end of the sentence.

Lines 108-109, 114-115: These sentences belong to the MM section.

Line 115-116: Please specify the index that was significant and indicate the p-value. Which statistical test was done to show the difference?

Line 124: Was there a statistical difference in relative abundances of archaea and viruses as you mention that the leaves of *O. meyeriana* had higher abundances than those in the other types of rice?

Line 132: Again, was there any statistical difference?

Line 139: Please indicate the LDA score.

Line 177: Please indicate the p-value.

Line 200: Please clarify what you mean by “global overview map” as a functional category in the metabolic category. My understanding is that this is a type of map that provides a high-level, overall view of metabolism, distinguished from regular metabolic pathway maps.

Line 206-207: Its quite interesting to see that a higher abundance of infectious disease i.e., bacterial and neurodegenerative diseases was detected in this study. Could this be of concern considering that this could be passed on to the seeds>

Line 219: Please indicate the number of KOs for each group.

Line 224: Again, “global and overview map” is not a functional category.

Line 252-253: This sentence belongs to the MM section.

Line 259: Does this apply to above ground communities or below ground communities? What about other factors such as the phyllosphere i.e., leaf surface microbiome, where bacteria can enter the plant tissues through stomata, wounds, or other natural openings, and potentially through seed transmission?

Line 263: Please provide a reference and “*Streptomyces caelestis*” should be italicized.

Line 265: Which are/including? I didn't see an

Line 277: This belongs to the MM section/ is already mentioned in the MM so there is no need to repeat in the Discussion section.

Line 278-281: This sentence is too long and needs to be split into two.

Line 352: How did the authors go about making sure that there was no contamination of the endophytic microbiome by the epiphytic “phyllosphere” microbiome when handling/processing the leaf material. This is not explained in addition to the total number/weight of leaves that was collected for each group.

Line 405: Apart from visualizing the communities using the PCoA which other statistical analysis was conducted to assess the difference between communities from the different wheat groups?

Line 407: Are the authors indicating that they used “Metastats” or they are just stating what it does as indicated in this statement “Metastats examines microorganisms or functions with significant differences between two groups”?

Title: Distinctive structure of endophytic microbial communities in two species of wild rice and 2 cultivated rice

Reviewer summary:

The authors of this study aim to build a base of knowledge about the microbiomes of leaves from different rice species including two wild and one domestic. They approach this characterization with the goals of describing species composition, comparing microbiomes across species, characterizing and comparing the functional profiles of these communities, and building a molecular network interaction model for the rice and their microbiomes. The subject matter is timely and important. Descriptive studies of plant microbiomes are important bases of knowledge for applied experiments and the data presented here is valuable for that reason. However, the study design is not sufficient to support many of the conclusions of the authors. For example, they do not seem to account for site analyses when that may be as or more important than host species in determining leaf microbiome structure. There is interesting data here, but the current manuscript needs extensive revision.

Comments by line:

Line 26-28 Inconsistent terminology. First “abundances” is used and then “relative abundance”. If you are referring to relative abundance please always qualify it as relative abundance.

Line 49-51 Fundamentally false. There are many endophytic pathogens. The source does not support this statement that endophytes “do not cause negative effects, injuries, or symptoms in their hosts”.

Line 51-52 This sentence is overly broad and the source does not support the statement that “most endophytes assist in plant growth”. First, endophytic microbiomes vary by plant species and many other factors. Second, demonstrating that the majority of endophytes in all plants contribute to “growth” (also a broad term) would require an incredible level of proof.

Line 57-59 Needs source. The statement “endophytes are the main factors that affect plant growth and development” is false.

Line 59-63 Not a comprehensive list. Rerword to say “some confirmed functions”

Line 66-67 Needs source.

Line 79-80 The source is about beets but the author is making a statement about rice. How is valuable defined in this context?

Line 80-82 Source does not support sentence.

Line 86 Questionable and simplistic as a universal statement. Sentence should state that this is the finding of one study.

Line 112-113 Source needed

Line 115-117 No description of how significance was determined. Needs test statistics and p values. All of this is also missing from the figure legend.

The entire results section needs more detail. Results are often reported qualitatively with no quantification. Many sentences refer to figures but do not have the figure referenced (e.g., lines 124-125 and 127).

Line 150 First reference to a random forest model. What is the model structure?

Lines 257-259 Needs source

Lines 261-263 Unsubstantiated. Data in this study were not sufficient to demonstrate that any associations between microbes and their host species pre-date rice domestication.

Lines 278-281 Confusing. I am not sure what this sentence is trying to say.

Lines 282-284 Needs source.

Lines 329-332 Sentences need sources.

The materials and methods section needs to be more detailed. Describe all models in detail. The metadata do not seem sufficient to draw many of the conclusions. For example, many of the differences in this study may be attributable to sire instead of plant species, however, study design and statistical methods do not seem to account for the effect of site. Other variables to consider incorporating include time of year, weather, and age of plants.

We thank the reviewers' insightful comments, which we addressed carefully point-by-point (in red) as below.

Reviewer #1 :

"The authors of the manuscript titled "Distinctive structure of endophytic microbial communities in two species of wild rice and cultivated rice" compared the endophytic communities of two wild rice and cultivated rice using a metagenomics approach and further provided theoretical basis for future utilization of endophyte in wild rice. The work is interesting and clearly motivated."

Response:

Thank you for your recognition of the overall research content of this article and the time you devoted in reviewing this article.

"Line 75: Please write "O". meyeriana in full at the beginning of the sentence."

Response:

We have corrected the writing according to your recommendation.

"Line 105: A full stop is missing at the end of the sentence."

Response:

We have added the period.

"Lines 108-109, 114-115: These sentences belong to the MM section."

Response:

We have removed the relevant content.

"Line 115-116: Please specify the index that was significant and indicate the p-value. Which statistical test was done to show the difference?"

Response:

We have conducted an analysis according to your instructions and added the relevant content to the text.

"Line 124: Was there a statistical difference in relative abundances of archaea and viruses as you mention that the leaves of *O. meyeriana* had higher abundances than those in the other types of rice?"

Response:

We have conducted statistical analyses and added the information to the revised manuscript.

Line 132: Again, was there any statistical difference?

Response:

We have conducted statistical analyses and added the relevant content, as per your suggestion.

Line 139: Please indicate the LDA score.

Response:

Thank you for the reminder. We have indicated all LDA scores in the manuscript.

Line 177: Please indicate the p-value.

Response:

We have conducted statistical analyses and added the relevant content.

Line 200: Please clarify what you mean by “global overview map” as a functional category in the metabolic category. My understanding is that this is a type of map that provides a high-level, overall view of metabolism, distinguished from regular metabolic pathway maps.

Response:

We have conducted a search in response to your question and found that KEGG classifies all metabolic pathways into multiple hierarchical tiers, where "Global and Overview Map" belongs to the top-level metabolic pathway classification (first-tier classification), representing a global overview of the metabolic network within organisms.

Line 206-207: It's quite interesting to see that a higher abundance of infectious disease i.e., bacterial and neurodegenerative diseases was detected in this study. Could this be of concern considering that this could be passed on to the seeds.

Response:

We propose that this result can be attributed to two factors. On the one hand, the KEGG pathway construction is centered on human diseases, leading to non-specific mapping of bacterial genes via sequence homology; on the other hand, the hypothesis of potential seed-mediated transmission (as previously discussed) cannot be dismissed, although this requires further validation in follow-up studies. Corresponding discussions have been added to the manuscript to address these considerations.

Line 219: Please indicate the number of KOs for each group.

Response:

We have added the number of KOs for each group.

Line 224: Again, “global and overview map” is not a functional category.

Response:

As mentioned for Line 200, KEGG classifies all metabolic pathways into multiple hierarchical tiers, where "Global and Overview Map" belongs to the top-level metabolic pathway classification (first-tier classification), representing a global overview of the metabolic network within organisms.

Line 252-253: This sentence belongs to the MM section.

Response:

The content you mentioned has been removed.

Line 259: Does this apply to above ground communities or below ground communities? What about other factors such as the phyllosphere i.e., leaf surface microbiome, where bacteria can

enter the plant tissues through stomata, wounds, or other natural openings, and potentially through seed transmission?

Response:

Thank you for your timely correction. The soil environment is indeed the primary influencing factor for rhizospheric endophytes, but this appears to be inapplicable to the phyllosphere. Revisions have been made considering your comment regarding "bacterial transmission through pathways such as stomata."

Line 263: Please provide a reference and "Streptomyces caelestis" should be italicized.

Response:

This perspective is primarily an inference derived from the study's findings; however, due to the initial failure to fully account for the limitations of these results, and considering reviewers' suggestions, we removed the aforementioned content.

Line 265: Which are/including? I didn't see an

Response:

We sincerely apologize for the ambiguity. The original intention was to identify wild rice-specific microbes through comparative analysis of endophytic communities across the three groups of samples as targets for follow-up research. However, after integrating comments from multiple reviewers, we have deleted this section and completely rewritten it to address concerns about research validity.

Line 277: This belongs to the MM section/ is already mentioned in the MM so there is no need to repeat in the Discussion section.

Response:

We have been removed the content you mentioned.

Line 278-281: This sentence is too long and needs to be split into two.

Response:

Thank you for your suggestion. We have split the long sentences into two shorter ones.

Line 352: How did the authors go about making sure that there was no contamination of the endophytic microbiome by the epiphytic "phyllosphere" microbiome when handling/processing the leaf material. This is not explained in addition to the total number/weight of leaves that was collected for each group.

Response:

Thank you for your feedback. The specific procedures for leaf disinfection treatment have been added in the corresponding section.

Line 405: Apart from visualizing the communities using the PCoA which other statistical analysis was conducted to assess the difference between communities from the different wheat groups?

Response:

The original intent was to use principal coordinate analysis (PCoA) for visualizing the results

of β -diversity assessments. I regret the earlier lack of clarity of the original text and have made corresponding revisions to the relevant content.

Line 407: Are the authors indicating that they used “Metastats” or they are just stating what it does as indicated in this statement “Metastats examines microorganisms or functions with significant differences between two groups”?

Response:

We used Metastats to examine microorganisms. The results are shown in Figure 2. I sincerely apologize for not making it clear. We have revised the text accordingly.

Reviewer #2 :

This study investigates the endophytic microbial communities in leaves of two wild rice species (*Oryza officinalis* and *O. meyeriana*) and cultivated rice (*O. sativa*) using metagenomic sequencing. Key findings include higher microbial diversity and network complexity in wild rice, identification of core microbiomes (e.g., Dechloromonas, Salmonella, Klebsiella), and enriched metabolic pathways in wild species. The authors suggest wild rice endophytes may offer beneficial traits for improving crop resilience and sustainability.

Response:

We greatly appreciate your comments regarding the research presented in this article and your dedicated efforts in reviewing the manuscript. Your valuable insights have significantly strengthened the overall quality of the work.

“The small Sample size (n=3 per group) limits statistical power. Please acknowledge this limitation and consider supplemental data (e.g., public datasets) to validate trends.”

Response:

Thank you for your valuable insight. We have duly addressed the limitations of our conclusions arising from the small sample size in the Discussion section. Additionally, we conducted an extensive search for relevant metagenomic sequencing data on the leaf endophytes of the two wild rice species investigated in this study within public databases; however, no applicable content was found at this time.

“Another issue is reliance on predictive KEGG annotations without metatranscriptomic/metabolomic validation. I suggest the authors to highlight this as a caveat and propose future experiments to confirm pathway activity.”

Response:

We thank the reviewer for noting the reliance on predictive KEGG annotations without metatranscriptomic/metabolomic validation. We acknowledge that KEGG-based pathway inferences are speculative and lack experimental confirmation. We explicitly framed this as a limitation in the revised Discussion, stating that functional interpretations require validation via metatranscriptomics or metabolomics. We also propose future studies incorporating these approaches to confirm pathway relevance. Thank you for strengthening our methodological rigor.

“What is the meaning of the presence of pathogens (*Salmonella*, *Listeria*) in core microbiomes without addressing contamination risks or host interactions? Discuss sterilization efficacy and potential plant-pathogen dynamics (e.g., commensalism vs. pathogenesis).”

Response:

We greatly appreciate your valuable suggestions. The relevant content has been added to the discussion section accordingly. “Core microbiome analyses detected pathogens such as *Salmonella* and *Listeria*. Surface sterilization methods aim to reduce contamination, but these taxa likely reflect endophytic colonization as some pathogens can enter plant tissues via stomata or wounds (46). A key limitation is that molecular methods cannot distinguish viable cells from non-viable ones; therefore, the detection does not confirm active infection (47). Ecologically, these bacteria may act as transient commensals in plants, using nutrients without causing symptoms, as it has been observed in crop endophytes (48). However, their presence suggests functional plasticity as the stress (e.g., wounding and nutrient stress) may trigger pathogenicity. Future studies using multiomics and inoculation assays are needed to clarify their roles.”

“Why do you focus on leaves that overlook root-associated microbes critical for nutrient uptake?”

Response:

We thank the reviewer for highlighting the significance of root-associated microbes in nutrient uptake. Our study focuses on leaf endophytes in wild rice primarily due to the fact that while existing studies have characterized the rhizospheric microbial community structure in wild rice, limited knowledge on its phyllosphere (leaf-associated) microbiome is available. Although we acknowledge the critical role of root microbiomes in nutrient acquisition and plant health, this study intentionally narrows its scope to characterize the community structure of leaf endophytes in wild rice, thereby aiming to fill this knowledge gap and provide complementary insights into the understudied aboveground microbial ecology of this species.

“Claiming about *Bradyrhizobium*'s role in leaves lack experimental support. Explain”

Response:

Regarding the role of *Bradyrhizobium* in leaves, we appreciate the comment that direct evidence for its functional mechanisms in this niche remains limited. Notably, in sections 322–328 of the original manuscript, we have explicitly discussed the uncertainty surrounding whether *Bradyrhizobium* performs nitrogen-fixing functions in leaves, emphasizing the need for follow-up experimental validation. Additionally, our literature review has identified studies reporting the presence of *Bradyrhizobium* in the phyllosphere microbiota. These findings indirectly corroborate the reliability of our research results, as they align with the detection of this genus in leaf-associated microbial communities described herein.

“Simplify visuals (e.g., merge panels, highlight key taxa) and clarify axis labels/legends.”

Response:

We have simplified the figure panels as per your suggestion, moved some images to the

supplementary figures, and added additional information.

“Although generally clear but requires proofreading for minor grammatical errors (e.g., inconsistent hyphens, tense shifts).”

Response:

Thank you for pointing that out! You are absolutely right, and we carefully proofread the text to address the inconsistent hyphens, tense shifts, and any other minor grammatical issues. Ensuring consistency and accuracy is crucial.

We have checked all the hyphens and en-dashes in the revised version. As for the tense shifts, please keep in mind that we used past tense to report our methods and results, present tense for published results, and also present tense when presenting the results of a figure. These tense shifts are the recommended ones.

“Specify p-values for PCoA/perMANOVA results and clarify significance thresholds in LEfSe.”

Response:

Thank you very much for your suggestions; we have added the relevant content.

“Overall, this manuscript provides valuable insights into the microbiome divergence between wild and cultivated rice, supported by robust metagenomic data. While methodological limitations exist, the findings lay a foundation for harnessing wild rice microbiomes to enhance crop resilience. With revisions addressing the above points, this study merits publication in Microbiology Spectrum.”

Response:

Thank you for recognizing the study’s value and robust metagenomic insights. All suggested revisions have been addressed. Thanks for your feedback, which helped us revise and strengthen the manuscript.

Reviewer #3 :

“The authors of this study aim to build a base of knowledge about the microbiomes of leaves from different rice species including two wild and one domestic. They approach this characterization with the goals of describing species composition, comparing microbiomes across species, characterizing and comparing the functional profiles of these communities, and building a molecular network interaction model for the rice and their microbiomes. The subject matter is timely and important. Descriptive studies of plant microbiomes are important bases of knowledge for applied experiments and the data presented here is valuable for that reason.

Response:

We are sincerely grateful for your recognition of the research presented in this article and your diligent efforts in reviewing the manuscript. Your invaluable insights have substantially enhanced the overall quality and rigor of the work.

“Line 26-28: Inconsistent terminology. First “abundances” is used and then “relative abundance”. If you are referring to relative abundance please always qualify it as relative abundance.”

Response:

Thank you for pointing that out. The term "abundances" has been changed to "relative abundance" accordingly.

“Line 49-51 Fundamentally false. There are many endophytic pathogens. The source does not support this statement that endophytes “do not cause negative effects, injuries, or symptoms in their hosts”.

Response:

Thank you for identifying the error. The relevant content has been revised, and a description of endophytic pathogens along with relevant references have been added.

“Line 51-52 This sentence is overly broad and the source does not support the statement that “most endophytes assist in plant growth”. First, endophytic microbiomes vary by plant species and many other factors. Second, demonstrating that the majority of endophytes in all plants contribute to “growth” (also a broad term) would require an incredible level of proof.”

Response:

We have revised the original statement to acknowledge the functional diversity of endophytes, citing specific evidence to avoid generalizations.

“Line 57-59 Needs source. The statement “endophytes are the main factors that affect plant growth and development” is false.”

Response:

Thank you for emphasizing the need for a source to support our statement and the accuracy of this claim. We acknowledge that the original assertion was overstated and has been revised, incorporating relevant references to strengthen the scientific rigor.

“Line 59-63 Not a comprehensive list. Reword to say “some confirmed functions”

Response:

Thank you for pointing out the error. We have revised it to "some confirmed functions".

“Line 66-67 Needs source.”

Response:

We have incorporated citations of relevant references to strengthen the scholarly basis of our arguments.

“Line 79-80 The source is about beets but the author is making a statement about rice. How is valuable defined in this context?”

Response:

Thank you for identifying the discrepancy. We have replaced the references with rice-specific studies that directly support our arguments, ensuring species relevance and evidential rigor.

“Line 80-82 Source does not support sentence”

Response:

We have revised the original content to: "In other words, during domestication, crops may lose genetic diversity and some host-specific endophytes, as it has been confirmed in wheat (24)." Corresponding references have been updated to recent wheat-focused research to strengthen relevance.

“Line 86 Questionable and simplistic as a universal statement. Sentence should state that this is the finding of one study.”

Response:

Thank you for identifying the error. The content has been revised to: "An early study first revealed the evolutionary relationship in wheat-microorganism interactions."

“Line 112-113 Source needed”

Response:

We have added relevant references.

“Line 115-117 No description of how significance was determined. Needs test statistics and p values. All of this is also missing from the figure legend.”

Response:

Thank you for pointing out this issue. We have conducted statistical analyses on all the mentioned contents and added ed them to the paper. We have also added citations to the figures.

“The entire results section needs more detail. Results are often reported qualitatively with no quantification. Many sentences refer to figures but do not have the figure referenced (e.g., lines 124-125 and 127).”

Response:

Thank you for the comment. We have added more detailed descriptions and conducted statistical analyses in the article, and also included citations to the figures.

“Line 150 First reference to a random forest model. What is the model structure?”

Response:

“Random forest is an ensemble-learning model made up of numerous decision trees. During construction, it randomly samples data and features, and finally integrates the results of all trees, thereby reducing the risk of overfitting and efficiently handling complex data.” This information has been added to the article.

“Lines 257-259 Needs source”

Response:

This perspective was originally a conjecture based on the study's findings. However, recognizing the initial oversight of the results' limitations and incorporating the reviewers' suggestions, we have decided to remove the aforementioned content.

“Lines 261-263 Unsubstantiated. Data in this study were not sufficient to demonstrate that any associations between microbes and their host species pre-date rice domestication.”

Response:

Thank you for promptly identifying the error. After comprehensive consideration of the manuscript's limitations, we have removed the content you mentioned.

“Lines 278-281 Confusing. I am not sure what this sentence is trying to say.”

Response:

We have deleted the mentioned content and completely rewritten this section according to your suggestions.

“Lines 282-284 Needs source.”

Response:

Thank you for the comment. We have added relevant references.

“Lines 329-332 Sentences need sources.”

Response:

We have added relevant references, based on your suggestion.

“The materials and methods section needs to be more detailed. Describe all models in detail.

Response:

Thank you for the suggestion. We have improved the Materials and Methods section.

“Many of the differences in this study may be attributable to site instead of plant species, however, study design and statistical methods do not seem to account for the effect of site.”

Response:

Firstly, we greatly appreciate your comment. The core argument of this paper primarily focuses on comparing the leaf endophytic communities among three different species under the same growth conditions. Through the results obtained and relevant references, we hypothesize that rice domestication might have resulted in the loss of certain endophytic microbial components. However, the initial manuscript might not have clearly conveyed this perspective. Therefore, in response to the comments from the three reviewers, we have comprehensively revised the manuscript.

Furthermore, we acknowledge the limitations in the experimental design and have added relevant discussions in the Discussion. Notably, the significance of this study lies in its preliminary metagenomic exploration and comparative analysis of leaf endophytes in the three rice materials, aiming to increase the limited existing understanding of these materials. In follow-up research, we will integrate metatranscriptomics, metabolomics, isolation culture,

and functional analysis to conduct more in-depth studies.

“Other variables to consider incorporating include time of year, weather, and age of plants.”

Response:

First, we appreciate your comment. In response, we have added information on material collection timing and weather conditions in the Materials and Methods section. The three types of plant materials used in the experiment were maintained in a greenhouse, minimizing the impact of the weather on the results. Additionally, the two wild rice materials are perennial, making plant age uncontrollable. To address this, we pruned the shoots and leaves of wild rice plants and collected leaves after new shoots emerged. The cultivated rice control materials were sown at the same time to maximize consistency across all samples.

Re: Spectrum02978-24R1 (Distinctive structure of endophytic microbial communities in two species of wild rice and cultivated rice)

Dear Dr. Zaiquan Cheng:

Your manuscript has been accepted, and I am forwarding it to the ASM production staff for publication. Your paper will first be checked to make sure all elements meet the technical requirements. ASM staff will contact you if anything needs to be revised before copyediting and production can begin. Otherwise, you will be notified when your proofs are ready to be viewed.

Sincerely,
Lindsey Burbank
Editor
Microbiology Spectrum